# Weak liquid water path response in ship tracks

Anna Tippett[1], Edward Gryspeerdt[1], Peter Manshausen[2], Philip Stier[2], and Tristan W. P. Smith[3]

[1]Department of Physics, Imperial College London, London, UK
[2]Department of Physics, University of Oxford, Oxford, UK
[3]UCL Energy Institute, University College London, London, UK

**Correspondence:** Anna Tippett (a.tippett22@imperial.ac.uk)

**Abstract.** The assessment of aerosol-cloud interactions remains a major source of uncertainty in understanding climate change, partly due to the difficulty in making accurate observations of aerosol impacts on clouds. Ships can release large numbers of aerosols that serve as cloud condensation nuclei, which can create artificially brightened clouds known as ship tracks. These aerosol emissions offer a "natural", or "opportunistic", experiment to explore aerosol effects on clouds while disentangling meteorological influences. Utilising ship positions and reanalysis winds, we predict ship track locations, collocating them with satellite data to depict the temporal evolution of cloud properties after an aerosol perturbation. Repeating our analysis for a null experiment does not necessarily recover zero signal as expected, but instead reveals subtleties between different null experiment methodologies. This study uncovers a systematic bias in prior ship track research, due to the assumption that background gradients will, on average, be linear. We correct for this bias, which is linked to the correlation between wind fields and cloud properties, to reveal the true ship track response.

We find that the liquid water path (LWP) response after an aerosol pertubation is weak on average, once this bias is corrected for. This has important implications for estimates of radiative forcings due to LWP adjustments, as previous responses in unstable cases were overestimated. A noticeable LWP response is only recovered in specific cases, such as marine stratocumulus clouds, where a positive LWP response is found in precipitating or clean clouds. This work highlights subtleties in the analysis of isolated opportunistic experiments, reconciling differences in the LWP response to aerosols reported in previous studies.

## 1 Introduction

A significant uncertainty in quantifying the effective radiative forcing (ERF) due to anthropogenic activity stems from the uncertainty in cloud responses to aerosol perturbations, known as aerosol-cloud interactions (Forster et al., 2020). The primary way in which aerosols can influence clouds is by acting as cloud condensation nuclei (CCN), thereby increasing cloud droplet number concentration ($N_d$) over very short timescales (Twomey, 1974). In the near instantaneous case, the water content of the cloud remains constant, therefore droplets become smaller on average (known as the Twomey effect; Twomey, 1977) and more reflective to incoming shortwave radiation, leading to a negative forcing on the climate's energy balance (a cooling effect; Forster et al., 2021).

However, over longer timescales, the water content of the cloud may change. Albrecht (1989) hypothesised that smaller droplets take longer to coalesce into rain droplets, implying that an aerosol perturbation would reduce precipitation efficiency

in a cloud (Rosenfeld, 2000). Consequently, this suppression of precipitation would enable a cloud to persist for a longer duration (the "lifetime effect") and result in an increase in the liquid water path (LWP) of the cloud. This increased water content, in turn, elevates the cloud albedo, leading to a negative ERF. Nevertheless, reduced droplet size can also promote the entrainment of dry air above the cloud, causing cloud desiccation, decreased LWP, and a warming effect (Ackerman et al., 2004; Bretherton et al., 2007). These are inherently time dependent processes, and attempts have been made to quantify the timescales over which these competing adjustments to clouds occur (Glassmeier et al., 2021; Gryspeerdt et al., 2021).

Previous studies have found a range of potential LWP responses to aerosols. Some studies, such as Small et al. (2009); Chen et al. (2012, 2014); Sato et al. (2018); Wall et al. (2022), suggest that the LWP will decrease following an aerosol perturbation. Others, such as Quaas et al. (2009); Koren et al. (2014); Grosvenor et al. (2017); Neubauer et al. (2017); McCoy et al. (2018); Rosenfeld et al. (2019); Gryspeerdt et al. (2021); Zipfel et al. (2022); Manshausen et al. (2022), argue that aerosols cause an increase in LWP in some conditions. Some studies, however, suggest that the LWP response will be weak (Malavelle et al., 2017) or bi-directional (Ackerman et al., 2004; Michibata et al., 2016; Toll et al., 2017, 2019; Gryspeerdt et al., 2019a; Possner et al., 2020; Glassmeier et al., 2021; Zhang et al., 2022; Fons et al., 2023). Typically, modelling studies suggest a uniform increase in LWP (Quaas et al., 2009; Michibata et al., 2016; Sato et al., 2018; Gryspeerdt et al., 2020), whereas observational studies are much more varied: large-scale studies typically find a LWP decrease (e.g. Chen et al., 2014), studies looking at the impact of effusive volcanic eruptions typically find no change to LWP (e.g. Malavelle et al., 2017; Toll et al., 2017; Gryspeerdt et al., 2019a), and other natural experiments such as ship track studies find both decrease/increases in LWP (e.g Christensen and Stephens, 2011; Toll et al., 2019; Christensen et al., 2022), depending on the situation.

The meteorological context in which the aerosol perturbation occurs is an important control on the sign of the LWP response, where it is typically suggested that LWP will likely increase in clouds that are clean and precipitating and decrease in clouds that are polluted and non-precipitating (Ackerman et al., 2004; Toll et al., 2017; Gryspeerdt et al., 2019a; Possner et al., 2020). The regional dependence of the LWP response, and therefore dependence on cloud regime also must be considered when comparing LWP responses between studies. Studies that investigate LWP responses to aerosols can often occur in different cloud regimes (marine stratocumulus, trade cumulus, etc.) which can have opposing responses (Lebo and Feingold, 2014). Any conclusion of the LWP response to an aerosol perturbation must be given in the context of the cloud regime and meteorology in which the study takes place.

Reports of the LWP response to an aerosol perturbation are varied, with different methods typically obtaining different effects. In order to reduce the uncertainty in our understanding of aerosol-cloud interactions, and any potential warming or cooling effects, it is important to reconcile these differences in the LWP response to aerosol perturbations. This will be vital for the assessment of the potential impacts of geoengineering (Feingold et al., 2024), as the conditions under which cooling could be induced remains a topic of uncertainty.

In this study, we investigate the LWP response to an aerosol perturbation, using ship tracks as our "natural experiment" to disentangle the meteorological covariance. Ship tracks refer to linear cloud formations often observed in the wake of ships, resulting from the release of aerosol particles into the cloud due to burnt fuel. By comparing the polluted cloud within ship tracks to the adjacent unpolluted clouds outside the tracks, one can isolate the aerosol effect on clouds (Conover, 1966; Durkee

et al., 2000). A review of the use of ship tracks as natural experiments can be found in Christensen et al. (2022). Moreover, ship tracks can be regarded as linear formations of independently perturbed clouds, as no information is transmitted along their length (Kabatas et al., 2013). This characteristic allows us to consider the distance along the ship track as a time axis, through which the cloud adjustment evolution after a perturbation can be determined (as in Gryspeerdt et al., 2021 and Manshausen et al., 2022). This previous work has demonstrated that the time evolution of the cloud response to aerosol is important to consider when investigating the sign and magnitude of the response (Glassmeier et al., 2021).

Many ship track studies utilise hand-logged track positions or employ automated track detection algorithms to identify polluted pixels in satellite imagery for analysis based on their appearance as quasi-linear albedo perturbations, either manually (Segrin et al., 2007; Christensen et al., 2009; Christensen and Stephens, 2011, 2012) or using machine-learning (Watson-Parris et al., 2022; Yuan et al., 2022). Manshausen et al. (2022) address the potential selection bias that these studies may have, as only the cloud response in visible tracks is considered. Recent work (Gryspeerdt et al., 2021; Manshausen et al., 2022) predicts ship track locations by advecting historical ship positions in reanalysis winds, thereby allowing a much greater number of tracks to be analysed.

The majority of these ship track studies split the cloud scene into clouds that are polluted (inside the ship track) and un-polluted by the ship emissions (outside the ship track). They then investigate the relative anomalies of cloud properties inside and outside the ship track in order to separate the aerosol effect from the covarying background meteorology. However, in doing so, these studies assume that the background gradients in the cloud properties will be linear, on average. This relies on the assumption that ship tracks are randomly oriented with respect background gradients in cloud properties, and therefore the "average" shiptrack will have a linear background gradient. This assumption is investigated in this work.

In this study, we establish the temporal development of $N_d$ and LWP in ship tracks in the Atlantic Ocean. As in Gryspeerdt et al. (2021), we use ship positions from transponder data (Smith et al., 2015), which are advected in three dimensions with ERA5 reanalysis winds (Hersbach et al., 2020) to predict ship track locations. Following Manshausen et al. (2022), we place no conditions on the ship tracks being visible in the satellite data and instead look at the combined effect of all visible and "invisible" tracks. We collocate these ship track locations with MODIS Aqua and Terra satellite overpasses (Platnick et al., 2017) to build up a composite image of the time evolution of cloud properties in ship tracks. To assess the impact of background cloud variation, we conduct a null experiment using ship locations from one year and cloud and wind data from a different year, effectively "sailing" the ships through the wrong year of wind and satellite data. We investigate any false signals seen in the null experiment composite, and by considering an alternative null experiment methodology, we isolate the cause of the false signal, revealing the importance of considering the background gradients in the cloud properties when analysing ship tracks. Using our correct null experiment to account for the natural covariability of clouds and winds, we isolate the causal aerosol impact on $N_d$ and LWP across the Atlantic. We investigate the conditions controlling the sign and magnitude of the response and use our corrected $N_d$ and LWP responses to place an estimate on the radiative forcing from LWP adjustments to changes in $N_d$.

## 2    Methods

### 2.1    Ship track location prediction

This work predicts ship track locations using a similar method to that of Gryspeerdt et al. (2019b, 2021), utilising over 35,000 ships from automatic identification system (AIS) transponder data in 2018, filtered to include specific ship types (large container vessels, bulk carriers, oil tankers, cruise ships and general cargo ships; Smith et al., 2015). The region of interest of this work is chosen to be the same as in Manshausen et al. (2022, 2023), to enable direct comparison of results. This region in the Atlantic Ocean bounded by (50º S,50º N) and (90º W, 20º E), and contains both stratocumulus and trade cumulus regimes.

We advect these ship locations forward in time for 36 hours using ERA5 reanalysis wind fields (Hersbach et al., 2020). This provides us not only with the predicted ship track location, but information about the time since that position of the ship track experienced the ship aerosol perturbation. Any errors in interpolation of ship location data from AIS will lead to incorrect ship track locations, therefore the resultant ship tracks are filtered to exclude cases where ships were moving unrealistically fast (with an apparent ship velocity of more than 40 knots). There will be some small additional uncertainty in this 'time since aerosol perturbation' in to the case where there is no relative motion between the ships and the clouds, however we estimate this to occur in a small number of cases.

#### 2.1.1    Vertical advection of ship plume

As a modification to the methods of previous studies that predict ship track locations, we impose vertical motion of the ship plume within our advection scheme. In the work of Gryspeerdt et al. (2021), the ship emission locations are advected using 1000hPa winds, thereby making an assumption of a constant plume height. Manshausen et al. (2022) aims to incorporate vertical motion by employing the HYSPLIT model (Stein et al., 2015), which relies on advection in the ERA5 vertical winds (Hersbach et al., 2020). However, these vertical winds are often close to zero, particularly in stratocumulus regions (due to low model resolution in ERA5), leading to minimal vertical rise of the resulting trajectories. In contrast, this research introduces a plume rise to the advection scheme, ensuring that the emission positions are advected at increasing heights along the length of the track. The plume rise equation used in this study is given by Briggs (1965):

$$H(t) = \left( \frac{3F_0 t^2}{2(1+k)\pi\beta^2 U_0} \right)^{1/3} \tag{1}$$

where, $H(t)$ is the height of the plume, $t$ is the time along ship track, $F_0$ is the buoyancy flux ($840\,\mathrm{m^4 s^{-3}}$), $\beta$ is the entrainment rate (0.3), $k$ is the added mass coefficient (1), and $U_0$ is the relative wind speed (using a representative value of $10\mathrm{ms^{-1}}$). Furthermore, the vertical motion of the ship track is capped at the boundary layer height from ERA5, ensuring that the plume is advected with the boundary layer, rather than higher-level winds.

## 2.2 Data

Ship positions are advected in ERA5 reanalysis winds at 0.25º resolution and 3-hourly intervals between the surface and the boundary layer top, which is also obtained from ERA5 (Hersbach et al., 2020). Cloud property data utilised in this study was acquired from NASA's Aqua and Terra satellites, equipped with the Moderate Resolution Imaging Spectroradiometer (MODIS). We locate our shiptracks in MODIS Aqua and Terra satellite imagery, leaving us with roughly 52,000 MODIS granules containing approximately 4,000,000 tracks. Cloud properties were extracted from the level 2 collection 6.1 dataset (MYD06L2 and MOD06L2, corresponding to Aqua and Terra, respectively; Platnick et al., 2017). To ensure data quality, a filtering process was applied based on the "Cloud_Multi_Layer_Flag", allowing only clear or single-layer cloud scenes, and restrictions on solar and sensor zenith angles (solar zenith angle < 65º and sensor zenith angle < 55º) were imposed to minimise potential retrieval biases (Grosvenor and Wood, 2014). To minimise the impact of the bowtie effect on pixel geolocation (Sayer et al., 2015), we regrid the MODIS data to 5km resolution. Additionally we filter our data to include only low-level clouds with cloud tops below 700hPa.

$N_d$ and LWP are calculated using MODIS effective radius and cloud optical thickness retrievals following Quaas et al. (2006); Grosvenor et al. (2018). Estimated inversion strength (EIS) was calculated from the potential temperature at 700hPa and the potential temperature at the surface (Wood and Bretherton, 2006), which were obtained from ERA5 reanalysis data. This data is only used to filter our ship tracks into different stability scenes in Section 3.2.2. The resolution of the ERA5 data is coarser than our central ship track region (roughly 25km), however we only consider the EIS values in the outside track region, therefore this should not be an issue.

## 2.3 Quantifying ship impacts on cloud

For each ship track, we investigate how cloud properties vary with perpendicular distance away from the center of track (in a similar method to that of Segrin et al., 2007). We define distance left of the shiptrack (with respect to the direction of travel of the ship at the head of the track) as negative, and right of the shiptrack as positive. Additionally, we use the associated time along the ship track to grid our MODIS data into 2D space - binning our cloud properties in time along and distance away from each ship track. This data is combined for all tracks to produce a "composite" ship track.

We define the polluted region inside the composite ship track as the region within 5km of the center of the track, and the clean outside region as the region 30-60km away from the center of the track. This "clean outside" region is assumed to be representative of the cloud properties at the track location, if there were not a ship track present. This means that we can isolate the aerosol impact on the clouds in the ship track, separating it from any changes in the surrounding meteorology.

We calculate the enhancement of cloud properties inside the track as the percentage difference between these polluted and clean regions. We define our enhancements in $N_d$ and LWP as $\epsilon_N$ and $\epsilon_L$, respectively. We calculate the enhancement from the composite ship track, rather than compositing individual enhancements in order to avoid errors (since the operations of calculating the mean of a distribution and calculating the ratio of two distributions are non-commutable; Manshausen et al., 2022). Errors on the enhancements are calculated using a bootstrapped method with 1000 samples (Efron, 1979).

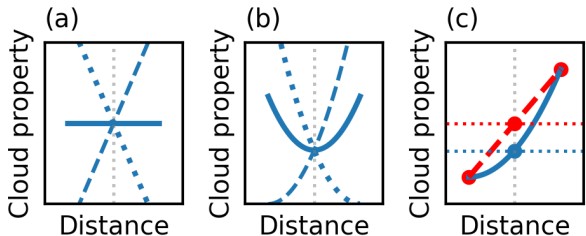

**Figure 1.** Subtleties in compositing cross-track background gradients in cloud properties. **(a)** Linear gradients will combine to form a linear composite background. **(b)** Non-linear gradients will combine to form non-linear composite backgrounds. **(c)** Any non-linearity in the composite background gradient will lead to false positive/negative enhancements in the composite ship track. In **(a)** and **(b)**, blue dotted/dashed lines represent the background gradients from individual ship tracks, and the solid blue line represents the composite background gradient when these gradients are combined. In **(c)**, the solid blue line represents the composite background gradient, and the red dashed line represents the average "outside" track value if a liner fit is assumed.

There are subtleties in the method used to combine all ship tracks into a composite ship track, which can significantly impact the calculated track enhancements. We summarise these subtleties in the following paragraphs.

Firstly, compositing every ship track means combining any background gradients in the cloud scene for each ship track. If the cross-track gradients are linear (i.e. the gradient in cloud property in the direction perpendicular to the ship track), when compositing all the ship tracks together, the composite track background will also be linear, and allow us to consider either side of the track equivalently. This is demonstrated in Fig. 1a, where combining linear trends will result in a linear composite. However, if the cross-track gradients are non-linear, then compositing all the ship tracks together will create a non-linear composite background (Fig. 1b) if the ship tracks are not randomly oriented with respect to the gradient.

Secondly, any non-linearity in the cross-track background gradient will lead to false positive/negative enhancements in the composite ship track (Fig. 1c). If the background gradient is concave (convex), then the average value outside the track will be greater (less) than the average value inside the track, even when there is no ship track present. This will lead to a false positive (negative) enhancement in the composite ship track.

## 2.4 Null experiments

In order to ensure any signal we see is due to the ship emissions and not a background effect, we repeat the same analysis for a null experiment. We require three components to conduct this ship track analysis: ship locations, reanalysis winds (to advect the ship locations and predict the track locations), and satellite data (from the advected ship track locations). When considering a null experiment, the correlations between these three components are important to consider, as subtle differences can bias results.

The null experiment chosen for this study uses the same ship locations as the real case (from 2018), but uses the winds and MODIS data from 2019. The resultant null experiment ship tracks predicted will most likely be incorrect, and not fall in the

|  | Ship locations | ERA5 winds | MODIS data |
|---|---|---|---|
| Real | 2018 | 2018 | 2018 |
| Null experiment | 2018 | 2019 | 2019 |
| Alternative (analogous to Manshausen et al. 2023) | 2018 | **2018** | **2019** |

**Table 1.** Sources of data for the cases analysed in this study. 2018 ship tracks uses ship locations from 2018, and winds and MODIS data from 2018. The null experiment of this study, referred to as 2019 uses ship locations from 2018, but winds and MODIS data from 2019. The alternative uncorrelated null experiment uses ship locations from 2018, but winds from 2018 to predict the track locations, and MODIS data from 2019.

same locations as any actual ship tracks, revealing any potential effects from our ship track orientations and the background gradients in the cloud properties and testing the assumption that ship tracks will be randomly oriented. Assuming the ship routes are only weakly constrained by weather conditions, this is equivalent to sailing our ships through a completely different year, and therefore the predicted ship tracks are very unlikely to align with any real tracks. There is a possibility of a small localised impact in any shipping corridors, but only when ship directions and winds are closely aligned. We expect this effect to be small in comparison to the total number of tracks.

We also investigate the sensitivity of our results due to the choice of year used for the null experiment by repeating our null experiment with 2017 data (see Fig. S5). We find that, whilst there is some interannual variability, it does not significantly impact the results of this study.

The null experiment methodology of this study differs from Manshausen et al. (2023), who consider a null experiment that uses ship locations and winds from a certain day to predict their ship track locations, but the satellite data from the day before (therefore the winds and satellite data will be uncorrelated). In this study, we retain the correlation between the winds and satellite data, as in the true ship track case this correlation will be present. Table 1 summarises the sources of data for the cases analysed in this study.

We calculate our corrected ship track response by calculating the difference between our real ship track case and the null experiment. This will remove any false enhancements due to background effects, and isolate the response of the cloud to the aerosol perturbation.

## 3 Results

### 3.1 Impact of null experiment choices

#### 3.1.1 Microphysical response

The time evolution of the $N_d$ and LWP enhancements are produced for up to 36 hours after the aerosol perturbation, for both the ship track case and the null experiment, and can be seen in Fig. 2a,b. We bin the $N_d$ data into 1 hour bins for the first 5

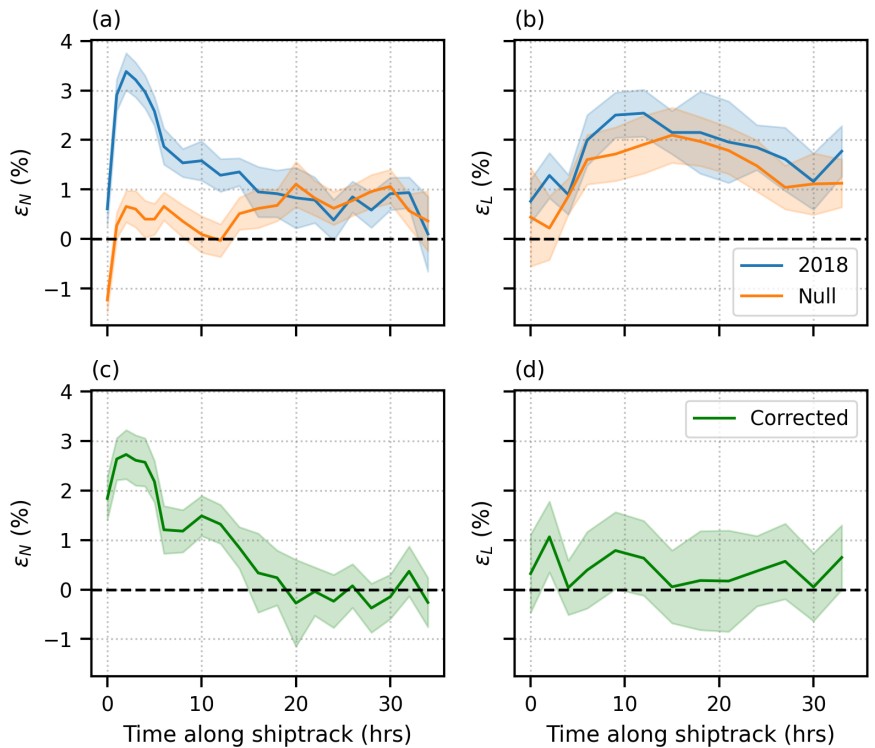

**Figure 2. (a)** $N_d$ and **(b)** LWP anomalies within ship tracks in 2018 as a function of time since aerosol perturbation, as well as background trend found in 2019 null experiment. 2019 null experiment uses ship locations from 2018, but winds and cloud data from 2019. LWP response in true ship track case and null experiment are found to be very similar in magnitude and time dependence. Corrected **(c)** $N_d$ and **(d)** LWP anomalies within ship tracks in 2018 as a function of time since aerosol perturbation. Responses are corrected by taking away the background signal, which is calculated from the 2019 null experiment.

hours, and then 2 hourly for the remaining time along track. Due to the noise in LWP data, we use 2 hour bins for the first 5 hours, then 3 hour bins for the remaining time.

The $N_d$ evolution is similar to those found in previous studies (Gryspeerdt et al., 2021; Manshausen et al., 2022, 2023), with

large increase in droplet number inside the ship track within the first 2-3 hours, and a decay back to the background state over the following 20 hours. The null experiment shows a constant enhancement in $N_d$ of roughly 0.5%, rather than recovering the null signal expected from the absence of any ship tracks. We observe a positive LWP anomaly that increases in magnitude for roughly 20 hours along the length of the ship track before decreasing. Surprisingly, we observe a very similar LWP response in the null experiment, with very similar evolution in time along the "track", despite the absence of any significant aerosol

perturbation (solely any small effect from shipping corridors).

The appearance of both a $N_d$ and a LWP enhancement inside the "ship track" region in the null experiment, despite the lack of any aerosol emissions, highlights a potential bias in previous work. Previous studies depend on the assumption that the clean

background cloud state can be identified by a linear average of the cloud conditions either side of the track when compositing millions of ship tracks. The presence of an enhancement in the null experiment suggests that this assumption may not be valid.

As previous studies (Manshausen et al., 2023) used a similar null experiment method to account for this effect and found no $N_d$ or LWP response, this discrepancy suggests that source of the bias lies in the method by which the null experiment is calculated. We isolate this bias, and its subtleties in the following sections.

### 3.1.2 Background gradients

The method by which enhancements are calculated involves taking an average of the cloud properties in the regions 30-60km away from the center of the composite ship track (on either side of the track), and calculating the percentage difference from the central 10km region. The purpose of the outside region is to estimate what the cloud properties at the track location would have been, if there was no ship track present.

However, if there is any non-linearity in the background gradient, then this will introduce an overestimation or underestimation of the actual value at the center, which will over/underestimate the signal in the center of the track, via the proposed mechanism shown in Fig. 1c. In essence, our estimate of what the cloud properties would have been at the track location if there was no ship track present will be incorrect, and therefore the enhancement calculated will be biased.

Fig. 3 demonstrates this for the null experiment (the case where no ship tracks are present) of this study. Since no ship tracks would be present in this experiment, we are solely seeing the impacts of the non-linear background gradients. In panels (a) and (b), the $N_d$ gradient in the composite is plotted at early times along "track" (between 0 and 5 hours), and at later times along track (between 15 and 20 hours). Panels (c) and (d) show the same, but for LWP.

The trends in the $N_d$ and LWP enhancements in the null experiment (orange lines in Fig. 2a,b), can be explained by how the non-linearity of the composite background gradient changes with time along track. The $N_d$ gradient is non-linear, thereby producing a small false positive enhancement in the center of the "track" (Fig. 3a). The non linearity of this gradient increases slightly but does not change significantly with time along track, therefore the false positive enhancement also only increases slightly with time along track (Fig. 3b), as seen in Fig. 2a. There is a small peak in $N_d$ in the center of the null experiment track, which can possibly be attributed to the presence of shipping corridors.

The LWP gradient, however, is slightly non-linear at early times along track, also producing a false enhancement (Fig. 3c), but becomes increasingly non linear with time along track (Fig. 3d), causing the magnitude of the LWP "enhancement" to increase with time (Fig. 2b). This null experiement reveals that the "enhancement" seen in the LWP response is actually a measure of the non linearity of the background LWP gradient, and not an aerosol effect.

The surprising similarity between the LWP response in the null experiment and "true" ship track case suggests that previous conclusions of LWP responses in ship tracks may be due to this false signal, rather than a true response to the aerosol perturbation. The increasing LWP in ship tracks observed in Gryspeerdt et al. (2021) and Manshausen et al. (2022, 2023) up to 20 hours is similar to the LWP responses observed in the null experiment and ship track cases of this study, and suggests that these previous studies may suffer from this bias. However, it is likely that any ship track study that calculates relative anomalies in a way similar to this study will suffer from this bias.

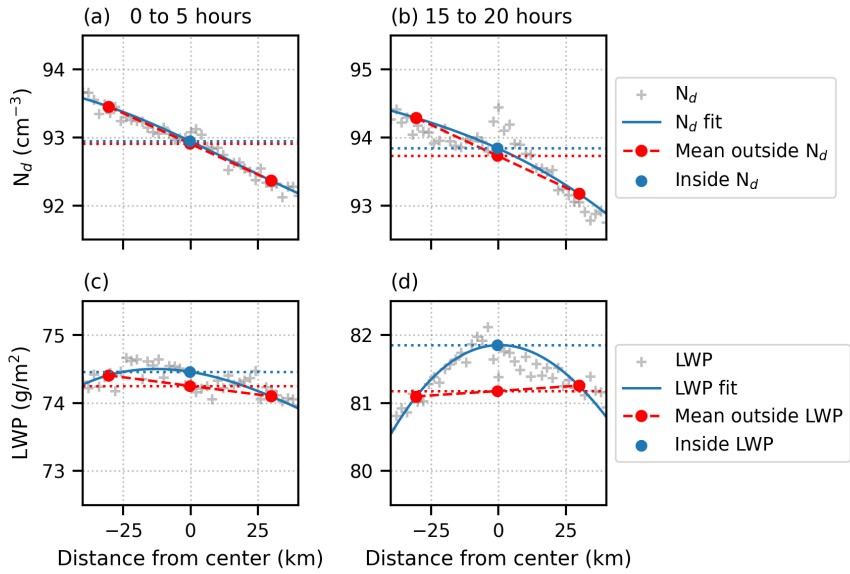

**Figure 3.** For our composite null experiment (with incorrect ship locations), we take a slice at early times along track (0-5 hours; panels **(a)** and **(c)**)) and later times along track (15-20 hours; panels **(b)** and **(d)**), and plot the observed $N_d$ and LWP as a function of distance from center of the track (grey crosses). In solid blue lines, we plot a polynomial fit (order 3) to the data, to demonstrate the non-linearity of the background gradients. In dashed red lines, we plot a linear fit calculated from the average "outside" track values (at 30km from the center of the track). The difference between the dotted blue and red horizontal lines represents the overestimation in the center of the track due to the non-linearity in the background gradient, hence a false positive enhancement. This false positive enhancement is relatively constant with time along track for $N_d$, but increases in magnitude for later times along track for LWP.

### 3.1.3 Correlations between LWP and wind fields

We attribute the source of this bias to the assumption that ship tracks are randomly oriented with respect to the background gradients in cloud properties, and therefore the composite ship track background gradient will be linear. This assumption will not be valid if there is a correlation between the ship track locations and the cloud property data retrieved. We investigate this by considering the correlation between the winds and the local maxima in LWP.

Fig. 4 shows the regional distribution of the LWP enhancement in the null experiment (where there should be no enhancements), and the correlation between the ERA5 winds and the second derivative in LWP (local maxima) in the Atlantic region. The metric for correlation is calculated by multiplying the second latitude (longitude) derivative with the wind speed orthogonal to the derivative direction, i.e. the zonal (meridional) wind speed component. We then take the length of the resulting two-component vector as the measure for correlation. For differentiation, we use the second order accurate central differences method implemented in numpy.gradient.

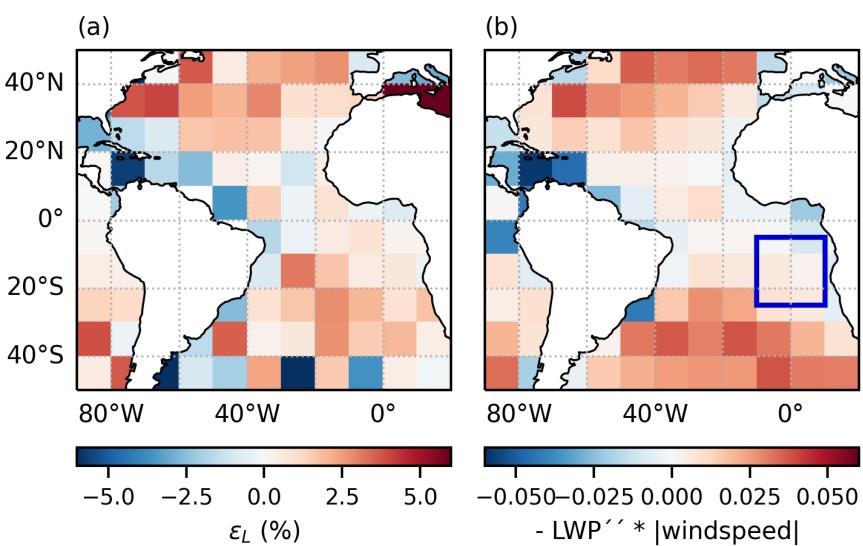

**Figure 4. (a)** Regional "enhancements" in the null experiment ship tracks, averaged over the 36 hour length of track and central track location binned to 10°. **(b)** Correlation between the second derivative in LWP (local maxima) and windspeed (from ERA5). The regional distribution of the LWP enhancement matches very closely to that of the correlation between maxima in LWP and winds, suggesting that this is the reason for non-linear background gradients in the composite. The navy box indicates the South East Pacific stratocumulus region investigated in Sect. 3.2

The regional distribution of the LWP enhancement in the null experiment matches very closely to the correlation between the winds and the local maxima in LWP, suggesting that this is the reason for the non-linear background gradients in the composite. We see the greatest false enhancements in the locations where the correlation between the winds and clouds is strongest, and therefore the composite background gradients are the most non-linear.

This result invalidates the assumption that averaging many ship tracks will produce a linear background gradient. Ship track locations are inherently a function of the winds in which they are advected, and therefore will be correlated to the clouds in which they are found.

Manshausen et al. (2023) do not observe LWP enhancements in their null experiment. In this null experiment, the ship positions and winds are from the same day, but the satellite data is from the day before. This means that there will be little correlation between the winds used to predict the track locations and the cloud properties retrieved, therefore when compositing

all the ship tracks, the cross-track gradients do average out to zero. This is in contrast to the null experiment used in this study, where the ship locations are from 2018, but the winds and satellite data are from 2019. We retain the correlation between the

winds and the cloud properties in our null experiment, and therefore reveal the bias due to the non-linear background gradients.

Repeating an analogous null experiment to Manshausen et al. (2023) (details can be found in Tab. 1), we find very weak LWP response (Fig. S1), further suggesting that the correlation between winds and cloud properties on a given day is the source of the bias. Additionally, we find very little correlation when we consider the mean winds and mean LWP maxima, highlighting the importance of considering daily correlations (see Fig. S2a) and individual weather systems.

This demonstrates the importance of correlations between cloud properties and winds, as all ship track studies will suffer from this bias when calculating enhancements inside track compared to unpolluted regions on either side of the track, regardless of the method used to predict the track locations, or whether the time dependence of the response is investigated. We only begin to see the significance of this effect when exploring the time evolution. At longer times along track, the ship track position is a greater function of the winds in which it is advected, and less dependent on the initial ship position. Thus, the

correlation between the cloud properties and the wind field becomes more significant. This can be seen in Fig. 2, where the LWP enhancement increases with time.

Whilst this effect will be present in all ship track studies that assume a linear background gradient, it will be much more significant in studies that consider all ship tracks, not just those that are visible. When considering all tracks, the ship track signal will be much smaller, and the dominant effect will be due to the non-linear background gradients.

It is worth noting that if we conducted our analysis by considering the enhancements in $N_d$ and LWP for each individual ship track, and then averaged these to obtain the composite (rather than compositing each ship track, and then calculating the enhancement), we would still see this false signal. This is because each individual ship track would still have a non-linear background gradient, and therefore the individual enhancements, whilst noisier, would still contain this bias.

### 3.2    Isolating the aerosol effect

We make the assumption that the non-linear background gradients in our null experiment are representative of this bias, and subtract the null experiment enhancement from the 2018 ship track enhancement to isolate the response of the cloud to the aerosol perturbation. The corrected $N_d$ and LWP responses can be found in Fig. 2c,d.

Comparing the 2018 responses and the corrected response, we see that the $N_d$ response remains largely similar in shape, but only a with a 3% enhancement in droplet number concentration after 2-3 hours. The LWP response, however, remains weak

(roughly 0.5%) for all times and shows very little evolution over time, as opposed to the strong positive LWP response seen in the uncorrected case.

Whilst the LWP response shows little changes over time, the sensitivity due to changes in droplet number ($\frac{\mathrm{d}\ln \mathrm{LWP}}{\mathrm{d}\ln N_d} = \frac{\ln \epsilon_L}{\ln \epsilon_N}$) will show some time dependence due to the time evolution of the droplet number perturbation, which is consistent with Glassmeier et al. (2021).

To investigate if we can observe a stronger LWP response, we filter our ship tracks into those that occur in polluted / clean backgrounds, stable / unstable environments, and precipitating / non-precipitating environments. We find that there is little

impact of these factors on the LWP when averaging across the entire Atlantic region, with the LWP response remaining noisy and close to zero for all times (see Fig. S3). This suggests that when averaging over all clouds in this large region, there is no control on the LWP response because so many clouds are insensitive to the aerosol perturbation.

Only when we consider a smaller subregion of the Atlantic, we recover a LWP response in certain conditions. We select a region bounded by (-15º N, 15º N) and (30º W, 0º W), which contains a large number of ship tracks in the marine stratocumulus deck in the South Atlantic (see box in Fig. 4b). This region is chosen as it contains a large number of ship tracks in a single cloud regime, and therefore we can investigate the controls on the LWP response in this regime. The results are presented in the following subsections, and in Fig. 5.

### 3.2.1    Background $N_d$

We subset our ship tracks into those that occur in polluted and clean backgrounds. We define the "outside" unpolluted region of each ship track as the distance between 30km and 60km away from the ship track, and calculate the average $N_d$ in this region. We then filter each ship track based in this background $N_d$. We consider those with background $N_d > 100$cm$^{-3}$ as polluted and those with $N_d < 50$cm$^{-3}$ as clean.

Fig. 5a,b shows the time evolution of the $N_d$ and LWP responses in polluted and clean background environments in a marine stratoculumus subregion. When considering this marine stratocumulus region, we find much greater enhancements in $N_d$ than seen in the entire Atlantic composite. We see greater maximum enhancement in $N_d$ in clean conditions (roughly 8%) than in polluted conditions (roughly 4%).

We find a non-zero LWP response in the marine stratoculumus region, with clean background clouds experiencing an in-
crease in liquid water content, and polluted clouds experiencing a slight decrease in liquid water content. This is consistent with there being a greater enhancement of entrainment in polluted regions, whereas the precipitation suppression mechanism is more dominant in clean regions, where there is more frequent drizzle to suppress due to smaller droplet number concentrations but greater droplet effective radii.

### 3.2.2    Inversion strength

Previous studies have suggested that boundary layer stability could potentially be a control on the strength of the cloud response to an aerosol perturbation (Toll et al., 2019; Possner et al., 2020; Manshausen et al., 2022). Using the estimated inversion strength (EIS) as a measure of atmospheric stability, we separate the ship tracks into those that occur in high EIS (> 3.5K, stable) and low EIS (< 3.5K, unstable) backgrounds, with Fig. 5c,d showing the time evolution of the $N_d$ and LWP responses in these different backgrounds. We use the same definition of "outside" track region as in Section 3.2.1.

We find that there is a weakly negative LWP enhancement in stable environments, and a weakly positive LWP enhancement in unstable environments. In both cases, however, the LWP response is weak and difficult to distinguish from the noise. Manshausen et al. (2022) found that there is a negative LWP anomaly in high EIS environments, and roughly zero LWP anomaly in unstable environments, whereas Toll et al. (2017) and Possner et al. (2020) find negative LWP responses in deeper boundary layers, which are commonly associated with lower EIS. We also see similar results to Manshausen et al. (2022) in the $N_d$

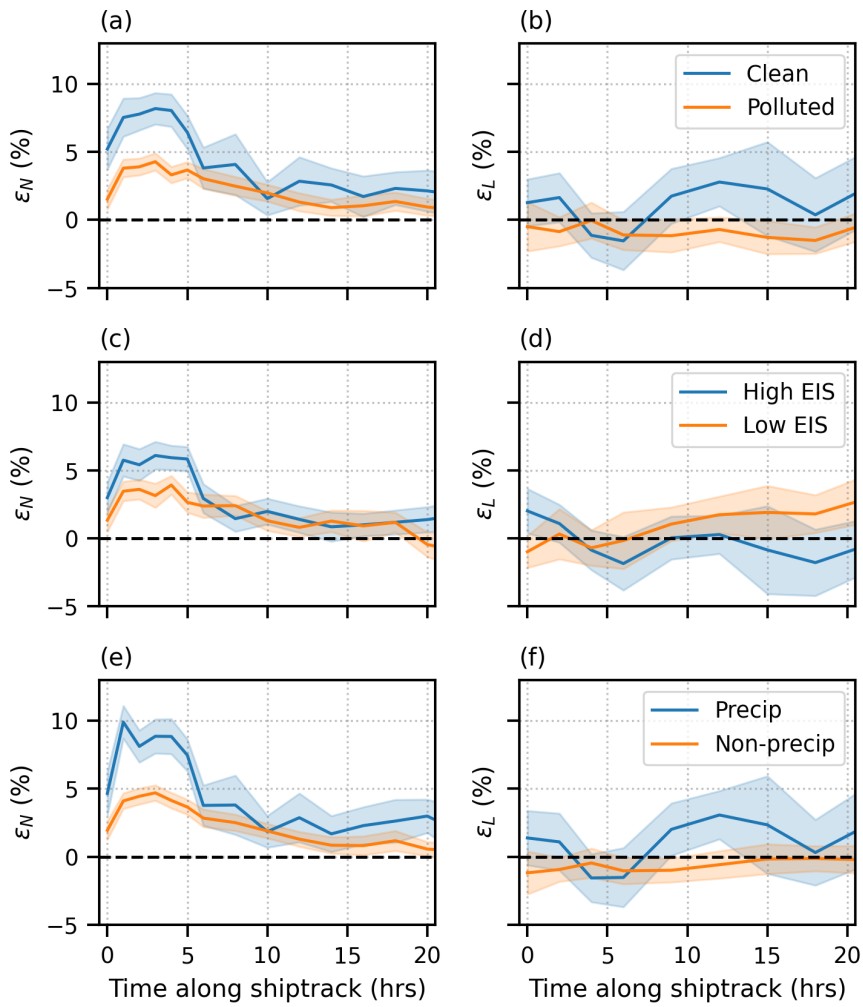

**Figure 5.** Time evolution of $N_d$ and LWP responses in (**a,b**) polluted and clean, (**c,d**) stable (high EIS) and unstable (low EIS), and (**e,f**) precipitating and non-precipitating background environments, for the marine stratocumulus subregion in the South Atlantic.

response, with a greater enhancement in droplet number concentration in stable environments than in unstable environments. This is consistent with stronger inversions occuring in shallower boundary layers and cleaner environments.

### 3.2.3 Precipitation

We define a precipitating background as one with average cloud effective radius (CER) greater than 15 $\mu$m, and non-precipitating as one with CER less than 15 $\mu$m (as in Toll et al., 2017). Fig. 5e,f show the time evolution of the $N_d$ and LWP responses in
these different backgrounds. Manshausen et al. (2023) require both inside and outside tracks to have CER $> 15\mu$m to define precipitating clouds, as cutting off the lower CER region of the distribution will lead to a bias in calculating the enhancements. We address this issue through the subtraction of the background signal from our null experiment, which would contain a similar bias and therefore the difference between the 2018 data and the null experiment should leave an unbiased signal.

We find that the $N_d$ response is greater in precipitating cases, with a 9% enhancement in $N_d$ after 2-3 hours, compared to a
340 5% enhancement in non-precipitating cases. The LWP response is positive in precipitating backgrounds, and weakly negative in non-precipitating backgrounds. This is consistent with the precipitation suppression mechanism - when background clouds are precipitating, this will be suppressed by smaller droplets on average being smaller (Albrecht, 1989) and causes an increase in LWP. The timescale for this onset appears to be roughly 6 hours. Wang and Feingold (2009a) observe an enhancement in LWP in clean clouds that is onset at roughly 5 hours after the perturbation, and is consistent with the LWP response clean and
345 precipitating clouds of this study. Gryspeerdt et al. (2021) find a faster LWP response to the $N_d$ perturbation (on the order of 2 hours), however it is likely this study suffers from non-linear background gradient bias identified in this work and therefore the timescales of this reponse are potentially inaccurate.

When background clouds are non-precipitating, there is no precipitation to suppress which may drive the slight decrease in LWP due to the enhancement in entrainment, as is consistent with the negative LWP in polluted regions (Fig. 5b). Manshausen
et al. (2023) find similar results, with a positive LWP response in precipitating clouds and roughly zero LWP anomalies for non-precipitating clouds.

### 3.3 Radiative forcing

Following the method of Manshausen et al. (2022), we calculate the sensitivity of LWP to $N_d$ for four equally sized EIS bins (defined in Table S1). We do not see EIS having a strong control on LWP response as was seen in Manshausen et al. (2022),
yet we elect to use the same method for the sake of consistency. We use our enhancements in LWP and $N_d$ that have been corrected for the background effect, by subtracting the null experiment response for each EIS bin.

We use all ship track observations from our region of interest (50ºN-50ºS and 90ºW-20ºE) to calculate these sensitivities, only using the cloudy ship track scenes of this study.

We calculate the sensitivities using $\frac{d\ln \text{LWP}}{d\ln N_d} = \frac{\ln \epsilon_L}{\ln \epsilon_N}$, where $\epsilon_L$ and $\epsilon_N$ being the corrected enhancements in LWP and $N_d$.
As in Manshausen et al. (2022), we calculate the LWP enhancement after 5 hours, and the $N_d$ enhancement before 5 hours, to provide an upper constraint on the potential cooling from the LWP response. The $N_d$ response is largest in the first 5 hours, and in Manshausen et al. (2022), the response plateaued after 5 hours. This meants that by using $\epsilon_N$ from the first 5 hours, and

$\epsilon_L$ from after 5 hours to calculate the sensitivities, we would calculate an upper estimate on the forcing. Using data from all times, the estimate of the forcing would become more negative, therefore this current method provides an upper limit.

We extrapolate these sensitivities globally to calculate an estimate of the global radiative forcing due to rapid adjustments in LWP, following the method of Manshausen et al. (2022); Bellouin et al. (2020). The global distribution of the sensitivity is found by taking into account the regional liquid cloud fraction (from MODIS), EIS (from ERA5), and our sensitivity of LWP to $N_d$ in each EIS bin. We use the estimation of $N_d$ changes due to aerosols from Bellouin et al. (2020). More detailed information on the forcing calculation can be found in Manshausen et al. (2022) and Bellouin et al. (2020).

In order to investigate if there is any control on the magintude of the forcing, we repeat this analysis with 2 and 12 equally sized EIS bins. This provides an estimate of the uncertainty in the forcing due to the choice of binning. We also investigate the sensitivity of the forcing due to the choice of year used for the null experiment (see Fig. S5), and find that the choice of year has little impact on the forcing estimate.

  We obtain an estimate of the forcing (and upper and lower bounds) of -0.16 (-0.29,-0.07) Wm$^{-2}$, which is weaker than
375 the estimate of -0.76 (-1.03,-0.49) Wm$^{-2}$ found in Manshausen et al. (2022). This is consistent with the false background enhancement contributing to an overestimation of the LWP response in ship tracks, and therefore also to the sensitivity of certain clouds to aerosol perturbations. Once we correct for this effect, we obtain much weaker LWP responses, and therefore weaker radiative forcing estimates. However, this result still suggests a cooling effect from the LWP response to aerosol perturbations, in contrast to the estimate of +0.2 (0.0,+0.4) Wm$^{-2}$ from latest IPCC report (Forster et al., 2021).

**4 Discussion and conclusion**

This work provides a better constraint on the response of clouds to an aerosol perturbation, and in particular the liquid water path (LWP) response and its effective radiative forcing. Following methodology similar to Gryspeerdt et al. (2021) and Manshausen et al. (2022), we use ship positions data and reanalysis wind fields to predict over 4,000,000 ship track locations in the Atlantic in 2018. From these, we investigate the time evolution of the $N_d$ and LWP in clouds after an aerosol perturbation.

Through the analysis of a null experiment, in which we "sail" our ships through the winds and satellite data of a different year, we identify a bias in ship track studies that causes an overestimation of the LWP enhancement in ship tracks. We suggest that the large positive LWP enhancements seen in trade cumulus ship tracks in Manshausen et al. (2022) are likely due to this bias, and that the LWP response to aerosol in these cases is much weaker.

  This effect can be attributed to the fact that non-linear cross-track background gradients in LWP do not average out to
390 zero when compositing many ship tracks, as they are not randomly oriented compared to the cloud field. We argue that the correlation between clouds and winds is the source of this bias. When considering an alternative null experiment which removes the correlations between ship track locations and cloud properties (analogous to the null experiment of Manshausen et al., 2023), we see that this LWP response disappears. This suggests that the correlation between winds and clouds is the source of this bias.

The subtle bias identified in this work will be prevalent in any ship track study that considers the relative anomaly of cloud properties inside the track compared to the unpolluted region on either side of the track. Despite this, in cases with a smaller number of visually verified tracks, the anomalies inside the tracks are likely to be much larger than the impact of this background effect, and therefore is unlikely to cause a change of sign of the response. Additionally, this bias is found to have a regional distribution, as seen in Fig. 4. The stratocumulus regions tend to have a much weaker bias compared to the cumulus

regions, therefore this bias is likely to be much less significant in studies that focus on stratocumulus regions.

This study predicts ship track locations with no requirement for tracks to be visible, and has track locations that are a strong function of the wind field. This is also the case in Gryspeerdt et al. (2021); Manshausen et al. (2022, 2023). In studies such as these, this bias becomes non-negligible due to the much weaker signal, the relative importance of weak tracks, and the significant correlation between cloud properties and the wind field in these locations. By correcting for this bias, we find that

the LWP response is close to zero in a composite of all tracks in the Atlantic region. This is in much closer agreement with LWP responses to the 2014 Holuhraun effusive eruption (Malavelle et al., 2017) and studies based on visible ship tracks (Toll et al., 2019). Chen et al. (2024) see slight decreases in LWP in a volcanic plume in various meteorological conditions in a trade cumulus regime, suggesting again that the LWP response to aerosol perturbations is weak, and not extremely positive as suggested by Manshausen et al. (2022).

We do find a LWP response when considering a subset of tracks in the Namibian stratocumulus deck. This suggests that cloud regime is an important control on the LWP response. It appears that the stratocumulus decks are much more sensitive to aerosol loading than shallow cumulus. Possner et al. (2020) suggests that the differences in LWP adjustments between shallow cumulus and stratocumulus are due to the lateral entrainment effects predominant in shallow cumulus, compared to the strong control on vertical moisture gradients and stability in stratocumulus.

We find an increase in LWP to aerosol in ship tracks that occur in clean, precipitating scenes, and negative LWP responses are found in polluted, non-precipitating conditions, in agreement with Ackerman et al. (2004); Gryspeerdt et al. (2019a); Toll et al. (2019). These results are consistent with the precipitation suppression mechanism in cleaner, precipitating clouds, in which there is precipitation to suppress via the decrease in droplet size. This enhancement through the precipitation suppression mechanism is seen at 5-6 hours after the aerosol perturbation, which is consistent with Wang and Feingold (2009a). These

results also support the idea that entrainment is enhanced more in polluted, non-precipitating clouds. The stability (EIS) is not found to have as strong a control on the $N_d$ or LWP response, with stable environments experiencing a weakly negative LWP enhancement, and unstable environments experiencing a weakly positive LWP enhancement, and $N_d$ enhancements being greater for more stable environments.

The results of this study are aligned with the findings of high resolution simulations of ship tracks. Wang and Feingold

(2009b) simulate ship tracks in a high-resolution model with double moment bulk cloud microphysics scheme and see very small changes to LWP in non-precipitating conditions. They do observe secondary circulation effects induced by the ship aerosol perturbation, however the composite nature of satellite observations in this study, and use of predicted ship track locations means we would be unlikely to observe this behaviour. Possner et al. (2015) also simulate ship tracks in a drizzling

stratocumulus deck and find that liquid water content was increased in some ship tracks. This is in line with our results in precipitating Namibian stratocumulus.

Using our corrected LWP and $N_d$ responses, we extrapolate globally to calculate an estimate of the radiative forcing from LWP adjustments. We find a weak, but negative forcing of -0.16 (-0.29,-0.07) Wm$^{-2}$ globally. This is much weaker than previously reported negative forcing estimates from ship tracks (Manshausen et al., 2022), and suggests that the LWP response to aerosol perturbations is closer to that determined from other lines of evidence (Malavelle et al., 2017; Toll et al., 2019).

Glassmeier et al. (2021) find that LWP adjustments in ship track studies can overestimate the cooling effect of aerosol perturbations when generalised to the global scale. This study avoids the issues suggested by Glassmeier et al. (2021) by considering ship tracks that are 20 hours long (on the order of the adjustment equilibrium time scale) and placing no requirement on tracks to be visible. However, the results of this paper do suggest an alternative way in which ship track studies can overestimate the LWP response to aerosol perturbations, and therefore their potential cooling impact, which must be taken into account when using ship tracks to investigate aerosol-cloud interactions.

The implications of these results are significant for the field of geoengineering. Marine cloud brightening (MCB) is often proposed as a method to mitigate the effects of climate change, by increasing the albedo of marine stratocumulus clouds through the injection of sea salt aerosol (Latham et al., 2012; Diamond et al., 2022). The sign of the LWP response, and hence the warming or cooling that an aerosol perturbation could induce, is vitally important to know with certainty in order to assess the effectiveness of MCB. Previous ship track studies (Manshausen et al., 2022), which suggest aerosol induced increases in LWP in ship tracks in shallow cumulus regimes, must be re-evaluated when considering the feasibility of MCB (Diamond et al., 2022; Hansen et al., 2023) since they will suffer from the bias identified in this study. This study hopes to emphasise the importance of the regional dependence of the LWP response, and the need for more studies in different cloud regimes in different meteorological contexts to fully understand the implications of MCB.

Although the magnitude, and time dependence of these response remain more uncertain, this study demonstrates the importance of the background environment in controlling the LWP response to aerosol perturbations, and emphasises the importance of considering non-linearities in the background gradients when interpreting enhancements from a background state. Once we consider these background effects, we find that the LWP response is very weak in a composite of all ship tracks in the Atlantic ocean in 2018, and that the marine stratocumulus deck LWP is much more sensitive to aerosol loading than shallow cumulus clouds. This reconciles the results of previous work, and provides a constraint on the radiative forcing due to LWP adjustments in clouds.

*Code and data availability.* MODIS data used in this work were acquired from Level-1 and Atmosphere Archive and Distribution System (LAADS) Distributed Active Archive Center (DAAC). The ERA5 data are from the Copernicus Climate Change Service (C3S) Climate Data Store (CDS). Ship AIS data was obtained from exactEarth. The code used in this work will be made available on publication.

*Author contributions.* AT and EG designed the study. AT performed the analysis, with PM contributing Fig. 4b. PM, EG, and PS assisted with the interpretation of the results. TS provided the ship locations data. AT drafted the manuscript, and PM, PS, and EG provided comments and suggestions.

*Competing interests.* At least one of the (co-)authors is a member of the editorial board of Atmospheric Chemistry and Physics.

*Acknowledgements.* AT and EG acknowledge funding from Horizon Europe programme under Grant Agreement No 101137680 via project CERTAINTY (Cloud-aERosol inTeractions & their impActs IN The earth sYstem), as well as a Royal Society University Research Fellowship (grant no. URF/R1/191602).

PM acknowledges funding from European Union's Horizon 2020 research and innovation programme under Marie Skłodowska-Curie grant iMIRACLI (agreement No 860100), as well as from the German Academic Scholarship Foundation (Studienstiftung des deutschen Volkes).

PS acknowledges support from UK Natural Environment Research Council project ACRUISE (NE/S005099/1), the FORCeS project under the European Union's Horizon 2020 research programme with grant agreement No. 821205 and the CleanCloud project under the European Union's Horizon Europe research programme with grant agreement 101137639 and its UKRI underwrite.

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
