# Peer review of "Weak liquid water path response in ship tracks"

_EGUsphere, 2024_

## Author Comment (AC1)

*We would like to thank the two anonymous referees for their helpful and insightful comments. Please find detailed responses to each individual comment below:*

**Referee 1**

**Specific comments:**

**It is a clever design that the study used null experiment to isolates the response of the cloud to the aerosol perturbation, and removes any effects due to the ship track geometries and alignment with non-linear gradients in the unperturbed. I wonder how representative is for the null experiment, since it was done with one particular year (2019) of meteorology and cloud images, for example, how much difference it could be for the correct LWP if choose 2017 for the null experiment instead of 2019? Some discussion of this uncertainty would help improve the robustness (and uncertainty range due to interannual variability of meteorology) of study and the new estimate of the corrected LWP adjustment.**

*This is a valuable comment and brings into focus the idea of what is the "correct" way to construct a null experiment that is representative of the 2018 meteorology. Due to interannual variability, we wouldn't necessarily expect 2019 data to be perfectly representative of the background meteorology in 2018, therefore we have performed an additional null experiment with the 2017 data to investigate the sensitivity on the choice of year used for the null experiment in our results. We have added Section 4 to the supplementary with the comparison between the 2017 and 2019 null experiments (see Fig. S5) and summarize the key points here.*

*We see largely the same LWP response shape in both null experiments, and when averaging the absolute difference between the two over the course of the 36 hours we obtain a difference of 0.3%. The $N_d$ responses in the two null experiments have a mean absolute difference of 0.5%.*

[Figure]

*Fig. S5*

*Using the 2017 null experiment to correct this 2018 ship track responses instead, and following the same procedure as outlined in the manuscript, we repeat our forcing calculation to give a sense of the uncertainty in this value due to the choice of null experiment year. We obtain a forcing estimate of -0.08 [-0.12,-0.07] $W/m^2$, suggesting that, as expected, there is a small sensitivity to the choice of null experiment year but does not significantly impact the results. The uncertainty range using the 2017 null experiment falls within that obtained from using the 2019 null experiment, therefore we retain the central estimate, applying a larger uncertainty interval. We have added discussion to lines 184-186 in Section 2.4 and lines 374-376 in Section 3.3 to discuss this extra analysis and sensitivity due to the choice of null experiment year:*

*L184-185: "...We also investigate the sensitivity of our results due to the choice of year used for the null experiment by repeating our null experiment with 2017 data (see Fig. S5). We find that, whilst there is some interannual variability, it does not significantly impact the results of this study."*

*L374-376: "...We also investigate the sensitivity of the forcing due to the choice of year used for the null experiment (see Fig. S5), and find that the choice of year has little impact on the forcing estimate."*

**It is nice that authors extensively discuss the new finding of this study compared against recent studies from satellite observations, e.g., (Manshausen et al., 2023; Manshausen et al., 2022); (Toll et al., 2017; Toll et al., 2019). But, it would also be nice to see some discussion of this new finding compared against modelling studies. For example, how does the corrected LWP compared against global climate models, and some high resolution simulations. On particular, (Glassmeier et al., 2021) used cloud-resolving simulation to also show that LWP adjustment could be overestimated by ship-track studies. Do you study confirmed their simulation with observational evidence, and how much agreement there is between your observation and their cloud-resolving simulation? Furthermore, in line-270 (ish), you find that LWP shows very little evolution over time, while Glassmeier et al. discuss that LWP adjustment would develop along the time (see their Fig.3). Does your new finding suggest that the cloud-resolving simulation also need significant improvement in the underlying fundamental processes?**

*Glassmeier et al., 2021 suggest that LWP adjustments will be overestimated by ship track studies since ship tracks are more likely to be sampled early in their lifetime, when they are more visible. These ship tracks will have only evolved for ~3 hours on average before sampling, which is a much shorter timescale than their proposed 20-hour adjustment equilibrium timescale. Our study uses ship tracks that are 20 hours long, and places no requirement on tracks to be visible, therefore aim to avoid this overestimation of the LWP response. This means that our generalisation to global estimates of forcings from LWP adjustments are still valid.*

*There are slight differences in the overestimations in Glassmeier et al., 2021 and our study. The overestimation in LWP response that our study reveals is an overestimation in the actual ship track response, whereas the overestimation in Glassmeier et al., 2021 is an overestimation that stems from when generalising ship track responses at short timescales to estimate global forcings from aerosols (at much longer timescales). We have clarified this by adding a paragraph in the discussion at line 439.*

*L439-444: "Glassmeier et al. (2021) find that LWP adjustments in ship track studies can overestimate the cooling effect of aerosol perturbations when generalised to the global scale. This study avoids the issues suggested by Glassmeier et al. (2021) by considering ship tracks that are 20 hours long (on the order of the adjustment equilibrium time scale) and placing no requirement on tracks to be visible. However, the results of this paper do suggest an alternative way in which ship track studies can overestimate the LWP response to aerosol perturbations, and therefore their potential cooling impact, which must be taken into account when using ship tracks to investigate aerosol-cloud interactions."*

*We see a good level of agreement with the results of Glassmeier et al., 2021, in that we see bi-directional responses in precipitating and non-precipitating background conditions respectively. The results in this work also agree well with the time-dependence shown in Glassmeier et al, 2021 – while there is little change in the LWP adjustment with time, the decreasing Nd enhancement would lead to an increase in the sensitivity, as seen in Glassmeier Fig. 3B. In the non-precipitating Sc case we would see the same negatively decreasing*

adjustment over time (using the equation adj = d ln LWP / d ln $N_d$ = ln $\epsilon_L$ / ln $\epsilon_N$ ). We discuss this in paragraph at line 295:

> *L295-297: "Whilst the LWP response shows little changes over time, the sensitivity due to changes in droplet number (d ln LWP / d ln $N_d$ = ln $\epsilon_L$ / ln $\epsilon_N$) will show some time dependence due to the time evolution of the droplet number perturbation, which is consistent with Glassmeier et al. (2021)."*

*This would suggest that at least some aspects of this work agree with high-resolution models. We have added in further discussion and comparison to high-resolution simulations of ship tracks at lines 428-434:*

> *L428-434: "The results of this study are aligned with the findings of high resolution simulations of ship tracks. Wang and Feingold (2009b) simulate ship tracks in a high-resolution model with double moment bulk cloud microphysics scheme and see very small changes to LWP in non-precipitating conditions. They do observe secondary circulation effects induced by the ship aerosol perturbation, however the composite nature of satellite observations in this study, and use of predicted ship track locations means we would be unlikely to observe this behaviour. Possner et al. (2015) also simulate ship tracks in a drizzling stratocumulus deck and find that liquid water content was increased in some ship tracks. This is in line with our results in precipitating Namibian stratocumulus."*

**It would be also nice to see some discussion of this new finding aligning with some recent satellite observation studies over stratocumulus and trade cumulus regimes, which is the focus of this study. For example, although (Malavelle et al., 2017) showed a negligible LWP adjustment using an Icelandic volcanic plume covering diverse cloud regimes, recently (Chen et al., 2024) used a Hawaii volcanic natural experiment over a trade cumulus regime and showed a slight but consistent decrease of LWP in various meteorological conditions.**

*Thank you for this comment, we have added discussion at line 411:*

> *L 411: "Chen et al. (2024) see slight decreases in LWP in a volcanic plume in various meteorological conditions in a trade cumulus regime, suggesting again that the LWP response to aerosol perturbations is weak, and not extremely positive as suggested by Manshausen et al. (2022)."*

**I can understand that the length of ship-track could be seen as a time-axis for ACI developing. However, I think this could only be true when the shipping routes are near-perpendicular to the prevailing wind. What about if they are near-parallel to each other, then the ACI signal could be a mixture of different time-scales? Would this influence your analysis, e.g. Fig.2 and Fig.5?**

*Predicted ship track locations are built up of trajectories from advection in wind fields. When shipping routes are near-parallel to the winds, ship track length will become much shorter in distance, however the distance between the "times" along the track will also become much shorter. This methodology does not directly convert distance to time, instead, we follow trajectories through time, therefore even when the shipping routes are parallel to winds, the "time along track" is still representative of how long since that segment of cloud experienced the ship aerosol perturbation. In the case where there is no relative motion between the ships and the cloud, the situation can occur where two 'times' occur within the same MODIS pixel, however*

*we estimate this to occur in only a small number of cases. To clarify this point of the methodology, and this small additional uncertainty, we have added the following to the paragraph at line 101:*

> *L101: "This provides us not only with the predicted ship track location, but information about the time since that position of the ship track experienced the ship aerosol perturbation [...] There will be some small additional uncertainty in this 'time since aerosol perturbation' in to the case where there is no relative motion between the ships and the clouds, however we estimate this to occur in a small number of cases."*

**Details of ERA5 data should be provided in Method. Would spatial resolution of ERA5, if I am correct would be around 25km, influence your analysis, give that your defined ship-track is about 10km central region?**

*At line 123 we have clarified specific details of the ERA5 data used in this study.*

> *L123: "Ship positions are advected in ERA5 reanalysis winds at 0.25º resolution and 3-hourly intervals between the surface and the boundary layer top, which is also obtained from ERA5 (Hersbach et al., 2020)"*

*When using ERA5 meteorological data (EIS) to filter our ship tracks into different scenes, we are only considering the background (outside track) value, therefore do not need to consider the central ship track region. Our outside region is greater than 25km, and therefore the resolution of ERA5 at 0.25º should be sufficient to obtain a value for the background EIS environment. We have added a sentence at line 137 to clarify this:*

> *L137-139: "This data is only used to filter our ship tracks into different stability scenes in Section 3.2.2. The resolution of the ERA5 data is coarser than our central ship track region (roughly 25km), however we only consider the EIS values in the outside track region, therefore this should not be an issue."*

**Editorial suggestions:**

**Removal the paragraph at line-160 (ish). Because it confused me when you show it here before explicitly introduce null experiment and tell about why, also you will talk about this point in the paragraph line-180, which is clearer.**

*Thank you for this comment, we agree that it is clearer to remove the paragraph.*

**Line 398: calling à cooling**

*Thank you, this has been amended now.*

**Referee 2**

**Specific comments:**

**I would expect that within 30-60 km (across track in the study is 30-60 km) cloud properties would be roughly similar on average. Such a distance is considered to be meso-scale and therefore I would not expect large variations due to varying synoptic conditions. What is the cause of the background gradients that you find across the tracks?**

**Why are both the null and 2018 LWP response to Nd are positive? It seems like there is some fundamental cause. In line 256, you write that it is the correlation between cloud properties and wind. What could be the mechanism causing this? Is it related to more surface fluxes at high wind speed?**

*Thank you for this comment, we have added a further Section 5 in the supplementary to discuss the source of these correlations and background gradients. We summarise the key points here.*

*Advected ship track locations are a strong function of winds, and therefore are a strong function of weather systems and the location of fronts. We typically find that within ship track locations we can see an imprint of the winds, as demonstrated in Fig. S6a. Since ship track locations are often aligned with these weather systems (Fig. S6b, bottom right corner), which themselves will consist of local maxima in LWP (Fig. S6c), when we consider the perpendicular-to-track gradient, we end up with our convex gradients.*

[Figure]

Fig. S6

**Figure 1 can be a bit confusing. The dotted lines in (a) and (b) are either crossing the midpoint (the track) or bouncing upward/downward. Consider choosing a different color or symbol.**

*Thank you for this comment, we have amended the figure and caption now to make it clearer.*

**Related to Figures 1 and 3, it is clear why concave (convex) matters, but if you have the observations at the track and on the sides, why do you look at the gradient from one side of the track to the other? Shouldn't it be enough from the center to one of the sides? And even so, you have the observations at the center (inside the track), so why do you interpolate over it? I assume that it's me not fully understanding the point, but others might too, and therefore I think it would be better to explain it more explicitly.**

*This analysis relies on calculating the aerosol impacts on clouds by comparing polluted ship track properties (inside the track region) to the surrounding clean clouds (which are assumed to be representative of the clouds at the location of the ship track if the ship emissions were not there. This hopes to remove any effects of surrounding meteorology and isolate solely the impact of the aerosols on the clouds. The purpose of the "outside" ship track region is to act as the reference state of the clouds, if there was no aerosol perturbation.*

*When we are dealing with ship tracks that are embedded within a region with background gradients in cloud properties, selecting this outside region becomes more complicated. If we were only to select one side of the track as the "clean" representative cloud, we would either be overestimating or underestimating what the properties would have been at the track location. Therefore, we must use information from both sides to gain an understanding of what the clean clouds would have been like.*

*Previous studies simply take a linear average of properties on either side of the track to estimate what clean clouds at the track location would have been, and then get the enhancement by calculating the* percentage *difference between the observed inside track properties, and these representative clean properties.*

*The purpose of Fig. 3 is to demonstrate that when background gradients are non-linear, taking a linear average of the outside regions to estimate what the unpolluted central track region would have looked like always results in an underestimation. This figure is using data from our null experiment – where there is no ship track signal present – therefore is aiming to demonstrate that the "enhancements" we see are a measure of the non-linearity, not any aerosol effect.*

*To make this point clearer, we have clarified the purpose of the "outside" background reference cloud properties in lines 147 and 219:*

> *L147: "This "clean outside" region is assumed to be representative of the cloud properties at the track location, if there were not a ship track present"*

> *L219: "…In essence, our estimate of what the cloud properties would have been at the track location if there was no ship track present will be incorrect, and therefore the enhancement calculated will be biased."*

*We have also attempted to clarify that the null experiment is revealing the non-linearity of the background, and not any aerosol effects in line 237.*

> *L237-239: "This null experiment reveals that the 'enhancement' seen in the LWP response is actually a measure of the non-linearity of the background LWP gradient, and not an aerosol effect."*

**The results are being compared to a few key studies which they are strongly related to. Nevertheless, I think it would be beneficial to expand the discussion to studies that used modeling as well. In addition, you mention that the time scale of the LWP response is an important factor. In this context, I suggest discussing the results with respect to Glassmeier et al., 2021 study (doi:10.1126/science.abd3980).**

*Thank you for this comment. In response to this, and comments from Referee #1, we have added further discussion and comparison to Glassmeier et al., 2021 in lines 439-444 and 295-297.*

> *L439-444: "Glassmeier et al. (2021) find that LWP adjustments in ship track studies can overestimate the cooling effect of aerosol perturbations when generalised to the global scale. This study avoids the issues suggested by Glassmeier et al. (2021) by considering ship tracks that are 20 hours long (on the order of the adjustment equilibrium time scale) and placing no requirement on tracks to be visible. However, the results of this paper do suggest an alternative way in which ship track studies can overestimate the LWP response to aerosol perturbations, and therefore their potential cooling impact, which must be taken into account when using ship tracks to investigate aerosol-cloud interactions."*

> *L295-297: "Whilst the LWP response shows little changes over time, the sensitivity due to changes in droplet number ($d \ln LWP / d \ln N_d = \ln \epsilon_L / \ln \epsilon_N$) will show some time dependence due to the time evolution of the droplet number perturbation, which is consistent with Glassmeier et al. (2021)."*

**In the radiative forcing calculation, is it 60°N-60°S or 90°N-90°S? Are you considering only scenes with the clouds you sampled and ignoring clear scenes and scenes with other cloud types when computing the mean forcing? It should be clear how exactly you derived the forcing.**

*We have added further details at lines 360 onwards to be more explicit about the methodology used to calculate the radiative forcing:*

> *L360: "We use all ship track observations from our region of interest (50ºN-50ºS and 90ºW-20ºE) to calculate these sensitivities, only using the cloudy ship track scenes of this study."*

**In line 358, you write that the cross-track gradients in LWP do not average out to zero. So why don't you calculate the gradients for individual ship tracks and then average them? You explain in Section 2.3 that this is done to avoid errors (which you calculate using a bootstrapped method). Maybe these errors are negligible compared to the averaging bias?**

*If we were to composite the enhancement for each ship track rather than calculate the enhancement from a composite ship track, we would be introducing significantly more noise and uncertainty to the response.*

*Individual ship track gradients are noisy. Calculating the enhancement involves combining data inside and outside the track, thereby increasing the uncertainty in this value. Combining many enhancements would cause this uncertainty to grow significantly. On the other hand, compositing the data for all tracks reduces the uncertainty in the composite track 2d space. Then we only calculate one enhancement from this, rather than millions.*

*Regardless, our gradients would not average out to zero in either method, as the non-linear composite originate from the fact that the ship tracks are not randomly oriented with respect to the background gradients.*

*We have added a sentence at line 283 to make this point:*

*L283: "It is worth noting that if we conducted our analysis by considering the enhancements in Nd and LWP for each individual ship track, and then averaged these to obtain the composite (rather than compositing each ship track, and then calculating the enhancement), we would still see this false signal. This is because each individual ship track would still have a non-linear background gradient, and therefore the individual enhancements, whilst noisier, would still contain this bias."*

**Minor comments:**

**Figure 4b: the label of the colorbar might has a typo.**

*Changed wind speed to windspeed*

**Line 283: "subsections" instead of "sections.**

*Thank you for pointing this out, this has now been amended.*

**Line 357: Clarify the meaning of "Nd enhancement before 5 hours". Why before?**

*We have clarified by rephrasing the sentence at line 364.*

**Lines 360-363: Sentence is not clear.**

*We have reworded and clarified this sentence at line 394.*

**Line 369: Seems like "be" is missing.**

*Fixed.*

**Line 398: "cooling" instead of "calling"?**

*Fixed.*